# Prevalence and mechanisms of high-level carbapenem antibiotic tolerance in clinical isolates of *Klebsiella pneumoniae*

Trevor Cross[1,2☯], Facundo Torres[1,2☯], Nadia Nikulin[1,2☯], Abigail P. McGee[2¤], Leena Jalees[2], Rhea Balakrishnan[2], Tolani Aliyu[2], Lars F. Westblade[3], Abhyudai Singh[4], Tobias Dörr (iD)[1,2,5,6]*

1 Department of Microbiology, Cornell University, Ithaca, New York, United States of America, 2 Weill Institute for Cell and Molecular Biology, Cornell University, Ithaca, New York, United States of America, 3 Departments of Pathology and Laboratory Medicine, Medicine, and Pediatrics, Weill Cornell Medicine, New York, New York, United States of America, 4 Department of Electrical and Computer Engineering, University of Delaware, Newark, Delaware, United States of America, 5 Cornell Institute for Host-Microbe Interactions and Disease (CIHMID), Cornell University, Ithaca, New York, United States of America, 6 Cornell Center for Antimicrobial Resistance Research and Education (CCARRE), Cornell University, Ithaca, New York, United States of America

☯ These authors contributed equally to this work.
¤ Current address: Graduate Program in Microbiology and Immunology, Geisel School of Medicine, Dartmouth, Hannover, New Hampshire, USA
* tdoerr@cornell.edu

## Abstract

Antibiotic tolerance is the ability of bacteria to survive normally lethal doses of antibiotics for extended time periods. Clinically significant Enterobacterales, for example, can exhibit high tolerance to the last-resort antibiotic meropenem. Meropenem tolerance is associated with formation of cell wall-deficient spheroplasts that readily recover to rod shape and normal growth upon removal of the antibiotic. Both the true prevalence of tolerance, and genetic mechanisms underlying it, remain poorly understood. Here, we find that meropenem tolerance is widespread among clinical Enterobacterales. Using forward genetics, we uncover tolerance factors in a hypertolerant isolate of the ESKAPE pathogen *Klebsiella pneumoniae*. We find that multiple mechanisms contribute to tolerance, and that cell envelope stress responses (PhoPQ, CpxPRA, Rcs phosphorelay and OmpR/EnvZ) collectively promote spheroplast stability and recovery, while the lytic transglycosylase MltB counteracts it. Our data indicate that tolerance is widespread among clinical isolates, and that outer membrane maintenance is a key factor promoting survival of tolerant *K. pneumoniae*.

**Data availability statement:** All relevant data are in the manuscript and its Supporting information files.

**Funding:** This work was supported by NIH grants R01AI143704 and R01GM130971 to TD. AS acknowledges support from NIH-NIGMS via grant R35GM148351. The funders had no role in study design, data collection and analysis, decision to publish, or preparation of the manuscript.

**Competing interests:** The authors have declared that no competing interests exist.

## Author summary

Bacteria can survive exposure to antibiotics by constantly repairing cellular damage induced by the drug. This "antibiotic tolerance" enables diverse bacteria to survive long enough to ultimately develop full antibiotic resistance. How they do that, and how widespread this phenomenon is in the clinic, is not well-known. Here we show that tolerance to a last-resort antibiotic, meropenem, is widespread in clinical isolates. Using one bacterium (*Klebsiella pneumoniae*, which can cause devastating pneumonia and bloodstream infections) as an example, we use a genetic screen to uncover genes that are required for tolerance. Such genes could encode targets for the future development of compounds that help antibiotics kill tolerant bacteria more effectively.

## Introduction

Healthcare-associated infections are common sequelae of hospitalization. Increasingly, treatment of these infections fails due to antibiotic resistance [1]. However, treatment also often fails despite the absence of outright resistance [2]. This has been attributed to other bacterial strategies for evading complete eradication, including the well-characterized phenomenon of bacterial persistence [3]. However, between the extremes of frank resistance and persistence, there is considerable complexity in bacterial antibiotic susceptibility patterns. For example, antibiotic tolerance, which has recently received increased attention as another means of surviving antibiotic insult, represents a state where a large fraction of susceptible bacteria survives in the presence of an antibiotic for extended periods, but does not proliferate. Gram-negative bacteria in particular can enter a non-replicative, cell wall-deficient spheroplast form upon treatment with β-lactam antibiotics [4–7]. This phenotype is reminiscent of L-forms [8–10] with the distinction that spheroplasts (unlike L-forms) do not proliferate in the presence of antibiotic. Spheroplasts are metabolically active, remain intact for extended time periods and readily recover their cell shape and resume division when the β-lactam drug is removed. Unlike in the case of heterotolerance (persistence) [3], spheroplast survival is afforded not by preventing antibiotic-induced damage through dormancy or reduction in growth rate, but rather by damage management through activation of stress responses [11–14]. While tolerant spheroplasts have been characterized in diverse Gram-negative pathogens [5–7], the true prevalence of β-lactam tolerance among Gram-negative clinical isolates remains unknown.

*Klebsiella pneumoniae* is a Gram-negative nosocomial pathogen that can cause a variety of opportunistic infections, including eponymous pneumonia, bloodstream infections and UTIs [15]. A member of the infamous ESKAPE group of pathogens, *K. pneumoniae* exhibits particularly worrisome levels of treatment failure [16]. Due to the high level of resistance against most β-lactams, the carbapenems remain one of the leading antibiotic classes that are still effective against *K. pneumoniae* [17].

Determining mechanisms that alter susceptibility against β-lactams in general, and carbapenems in particular, is thus a critical need for developing new treatment options against *K. pneumoniae.* β-lactams covalently and irreversibly bind to so-called penicillin-binding proteins (PBPs), which are the principal synthases of the main cell wall component peptidoglycan (PG). β-lactam binding results in several downstream events that contribute to cell death in poorly-defined ways [13]. Endogenous cell wall lytic enzymes (collectively referred to as autolysins) degrade PG into monomers and ultimately eliminate the sacculus entirely. In some bacteria, cell wall degradation can cause catastrophic rupture and cell lysis, for example in the well-studied model organism *E. coli* [13]. In addition, the constant synthesis and immediate autolysin-mediated degradation of PG (futile cycling) depletes cellular resources, potentially exacerbating detrimental effects independent of cell lysis [18,19]. Interestingly, β-lactam exposure coincides with the generation of damaging reactive oxygen species (ROS), though the contribution of ROS to β-lactam killing is not entirely clear [11]. It is thought that the combination of these direct and indirect effects collectively contributes to cell death by lysis and/or internal damage. How tolerant cells manage this assault to stay alive has remained poorly understood, but previous work has implicated cell envelope stress signaling in spheroplast survival [11,12,20]. In the cholera pathogen *Vibrio cholerae*, for example, the VxrAB system is induced by β-lactam stress, resulting in the upregulation of cell wall synthesis functions (which likely readies the cell for recovery once the antibiotic is removed) and downregulation of detrimental iron influx (putatively to reduce the damage sustained by ROS production) [11].

In this study, we sought to characterize meropenem tolerance in *K. pneumoniae*. We found that clinical isolates of this nosocomial Gram-negative pathogen often exhibit high levels of meropenem tolerance. Using TnSeq, we defined genetic requirements for tolerance, and uncovered a role for cell envelope stress signaling systems in both spheroplast maintenance and recovery. Our data open the door for a more in-depth understanding of the molecular mechanisms promoting tolerance, priming the future design of antibiotic adjuvants that eradicate tolerant cells.

## Results

### *Klebsiella* species clinical isolates exhibit high meropenem tolerance

As part of a larger effort to determine the role of antibiotic tolerance in healthcare settings, we characterized a large panel of bloodstream isolates from the culture collection of Weill Cornell Medicine (WCM). This de-identified 271 isolate panel (S1 Table) has been fully characterized through antimicrobial susceptibility testing and consists of Gram-negative pathogens that are mostly carbapenem-susceptible (MIC < 1 μg/mL), with three isolates testing as intermediate (MIC = 2) and seven strains as fully resistant (MIC > 4 μg/mL; one strain due to its possession of a KPC carbapenemase). Specifically, the panel consists of 52% *Escherichia coli* isolates, 30% *Klebsiella pneumoniae*, 9% *Pseudomonas aeruginosa*, 6% *Enterobacter cloacae* complex, 2% *Klebsiella variicola* and 1% *Acinetobacter baumannii* complex. Since the majority of the isolates are carbapenem susceptible, we considered this an ideal panel for determination of how prevalent carbapenem tolerance (rather than resistance) is in clinical samples. Currently, no fully-established sufficiently high throughput tolerance assay exists (we considered another established tolerance test, the TDtest [21] as less effective for our purposes due to the high number of plates required), and we therefore chose to use a semi-quantitative assay for our screen. To this end, we grew the isolate panel (curated to remove the resistant isolates with MIC > 4 μg/mL, with the exception of the KPC producer as an example of full resistance) in a 96-well format, where we exposed cells to meropenem (10 μg/mL, 5x – 100x MIC) at high cell densities (following a 10-fold dilution of overnight culture into fresh medium) in Brain-Heart Infusion broth. At the 8 hour and 24 hour timepoints, 5 μL of thus-treated cultures were then spotted on an agar plate together with 5 ng of purified carbapenemase (KPC-2) to remove the antibiotic. After 24 hours of incubation, spots were then scored as full growth (a confluent spot = high tolerance), growth with a few visible colonies (= intermediate) or no growth (= low tolerance) (Fig 1A and S2 Table). Since this panel contained isolates exhibiting a wide distribution of susceptibilities to meropenem (MICs ranging from ≤0.015 – 4 μg/mL), we considered the possibility that apparent differences in tolerance may instead reflect differences in MIC. Plotting tolerance categories over MIC values revealed that all

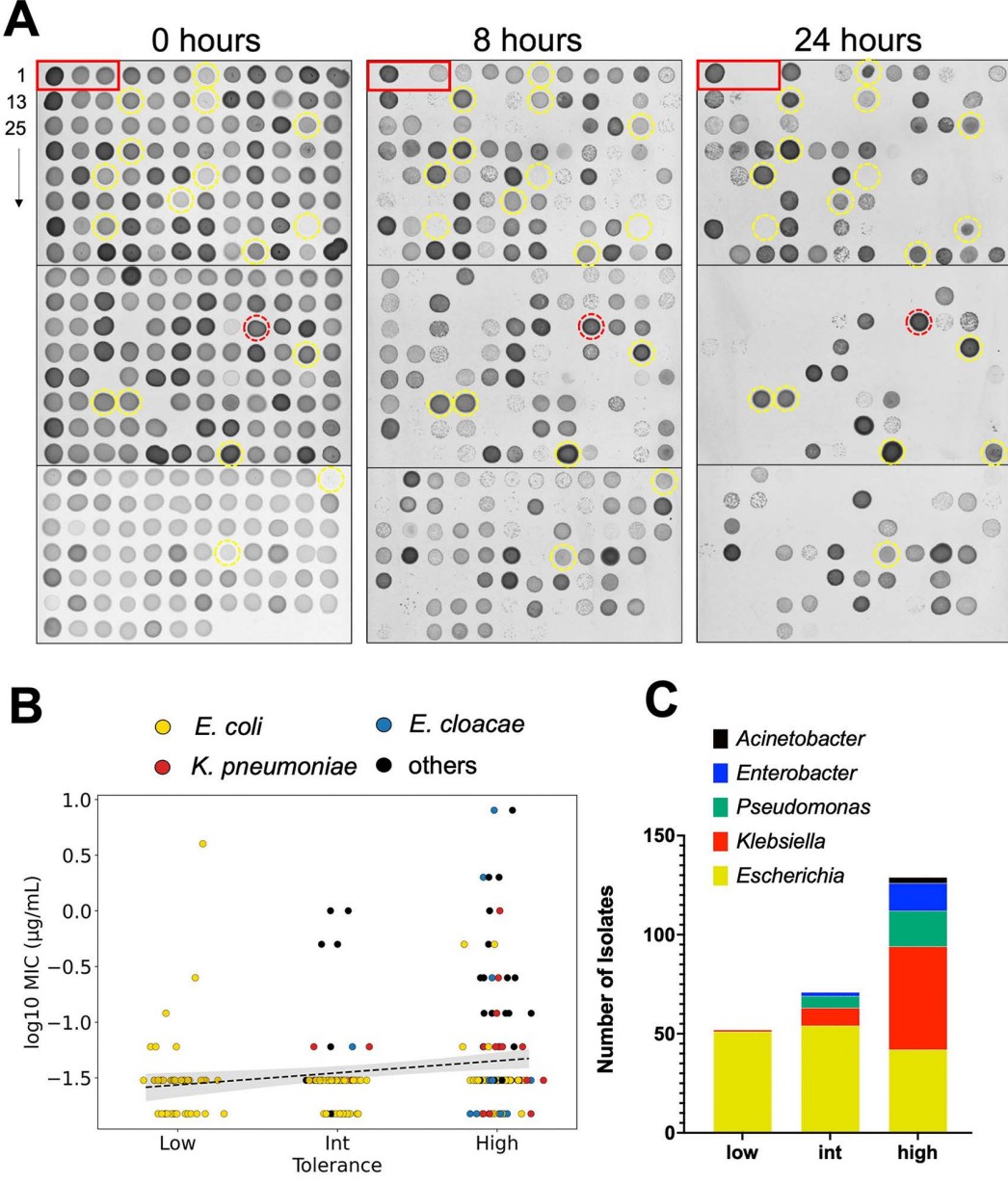

**Fig 1. Widespread meropenem tolerance among bloodstream isolates. (A)** A panel of susceptible bloodstream isolates (meropenem MIC < / = 4 µg/mL, n = 263, see text for details) was exposed to 10 µg/mL (>2x MIC for all isolates) meropenem in 96-well format (BHI medium) and incubated for 8 hours or 24 hours. Following incubation, purified KPC-2 enzyme was added to the wells, and 10 µL transferred to a BHI plate without antibiotic. Tolerance was scored qualitatively as the ability to form colonies/confluent growth after 8 or 24 hours. Yellow circles indicate examples of different tolerance levels across high MIC (>/= 1 µg/mL). Red circle indicates a positive control isolate (resistant, KPC+). Red box indicates examples of (from left to right) a highly tolerant *Klebsiella pneumoniae* isolate, an intermediate tolerant *E. coli* and a low tolerant *E. coli*. Shown is a representative panel from 3 biological replicates **(B)** Tolerance categories were plotted with their corresponding MIC values). Black line indicates weak correlation between tolerance and MIC (Spearman's coefficient = 0.22); grey area represents 95% confidence interval. Raw data are color-coded according to species **(C)** Genus distribution among the tolerance categories.

3 tolerance categories contained isolates with similar variance in MICs (Fig 1B). As an example, all isolates in Fig 1A with a yellow circle had comparatively high MICs (>/= 1 µg/mL), but displayed very different survival levels at 8 and 24 hours of meropenem exposure. While we did observe a slight and statistically significant positive correlation between tolerance category and MIC (Spearman coefficient $\rho = 0.22$, $p = 3.72^{-4}$) (Fig 1B), the large overlap in MIC between categories and the weakness of the correlation suggest an insignificant relationship between tolerance and intrinsic resistance. Thus, tolerance as determined in this assay is not a simple function of MIC variation; rather, these data suggest that our assay largely captures true tolerance.

We then asked whether some genera were more likely to exhibit tolerance than others, and to this end plotted tolerance as a function of bacterial genus. Interestingly, the low tolerance fraction was almost exclusively occupied by *E. coli* isolates (consistent with our previous, qualitative observations [7]), while in the high tolerance fraction, *Enterobacter species*, *Pseudomonas aeruginosa*, and especially *Klebsiella pneumoniae* were highly represented (Fig 1C and S2 Table). Statistical analysis confirmed a strong enrichment for *E. coli* in the low tolerance category (odds ratio = 59.94, p = 5.2x10$^{-13}$), and enrichment for *K. pneumoniae* in the high tolerance category (odds ratio = 7.18, p = 8.4x10$^{-9}$). Thus, *K. pneumoniae* clinical isolates in this panel are more likely to display antibiotic tolerance compared to *E. coli*.

## The *K. pneumoniae* TS1 isolate is highly meropenem tolerant

Since *K. pneumoniae* clinical isolates answered our survey as particularly tolerant, we chose to focus on this species for further study of meropenem tolerance. However, we noted that different *K. pneumoniae* isolates diverged in tolerance levels in our spot assay (Fig 1A). To more quantitatively determine strain-specific differences, and to identify new model strains for mechanistic studies, we compared two well-characterized clinical isolates of *K. pneumoniae* for their tolerance levels *in vitro*, namely *K. pneumoniae* strains TS1 (a derivative of the CDC isolate bank AR0080) and KPNIH1 [22]. Both strains have been cured of their carbapenemase-containing plasmids and are thus fully meropenem susceptible with MIC's of 0.05 (KPNIH1) and 0.0625 - 0.1 (TS1), respectively (S3 Table). Time-dependent killing experiments in the presence of 10 µg/mL meropenem revealed substantial differences in tolerance levels between the two strains, despite their similar susceptibility (Fig 2).

This observation supports our previous finding of often different tolerance levels between different isolates of the same species [7]. Both *K. pneumoniae* strains still exhibited higher levels of survival relative to the low-tolerance *E. coli* MG1655 comparison strain, which exhibited a steep, 4.5 log reduction in viability within six hours of meropenem treatment, and complete elimination by 24 hours of exposure. The TS1 strain was almost completely tolerant, with only a slight (5–10-fold) decrease in viability even after 24 hours of exposure (Fig 2A). In contrast, the KPNIH1 isolate lost viability by 10,000-fold after 24 hours. In *both Klebsiella* isolates, survival coincided with formation of cell wall-deficient spheroplasts (Fig 2B) that readily recovered to rod shape morphology upon removal of the antibiotic (Fig 2C). We also used these three benchmark strains to validate our semi-quantitativetolerance spot assay (Fig A in S1 Text). Consistent with the time-dependent killing experiments (Fig 2A), TS1 scored as highly tolerant, KPNIH1 as intermediate, and MG1655 as non-tolerant in the spot assay (Fig A in S1 Text). Thus, the TS1 isolate is highly meropenem tolerant, and serves as an ideal model for a mechanistic characterization of spheroplast formation and recovery.

## A TnSeq approach reveals putative tolerance genes in *K. pneumoniae* TS1

We next sought to identify factors that contribute to meropenem tolerance in *K. pneumoniae*. To this end, we conducted a transposon-insertion sequencing screen to assemble a genome-wide mutant fitness map in the presence of meropenem and compared insertion frequencies before and after 6 hours of meropenem exposure (including an outgrowth step, see methods for details, Fig 3A).

The screen uncovered both expected and novel putative tolerance genes (Fig 3B and S4 Table). Notably, the gene coding for PBP1b was detected within the top two percent of candidate essential genes for tolerance to meropenem.

PLOS Pathogens

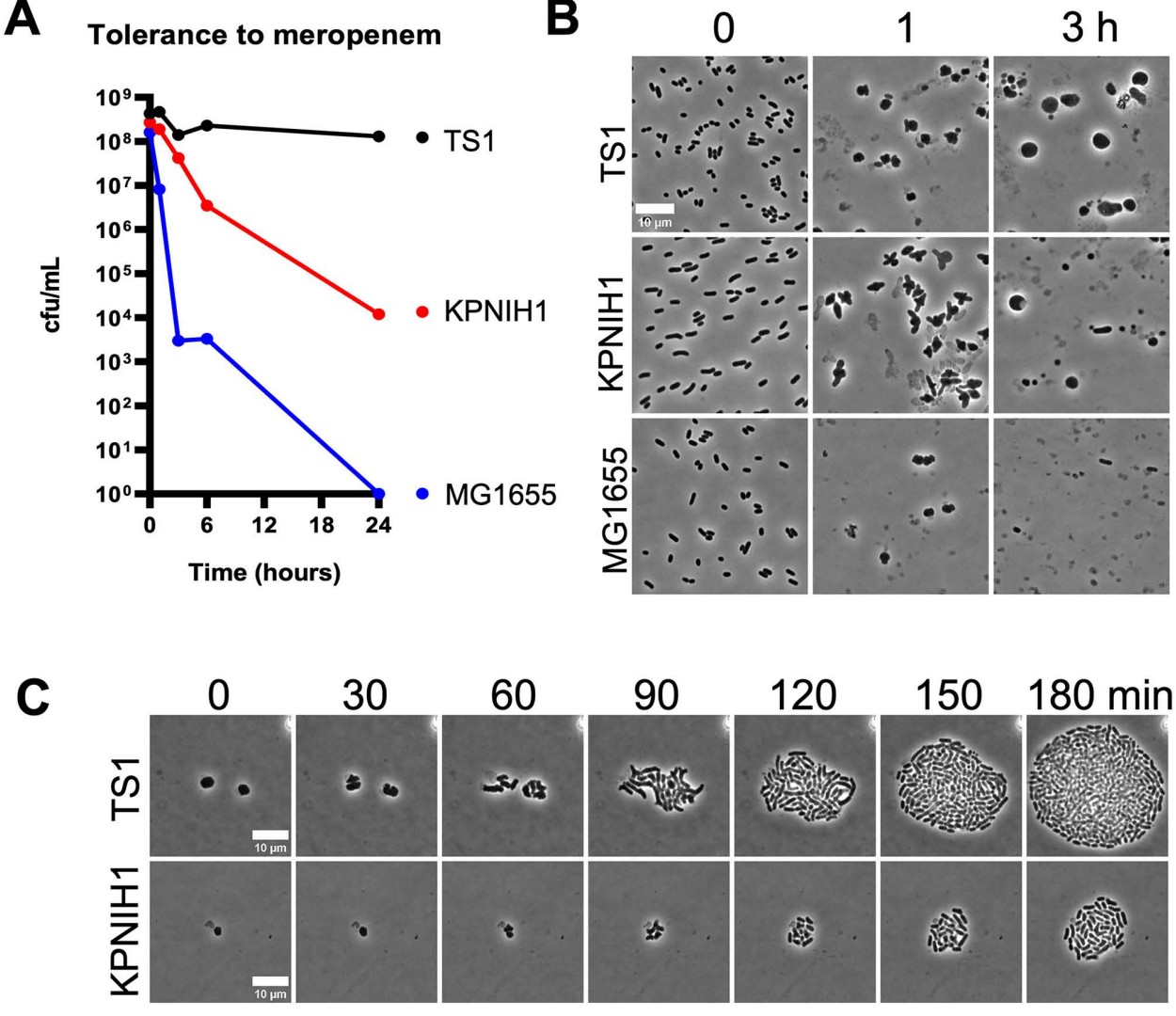

**Fig 2. *K. pneumoniae* TS1 is a hypertolerant clinical isolate. (A)** Example time-dependent killing experiment (BHI medium) in meropenem (10 μg/mL) reveals differences in tolerance levels among two *K. pneumoniae* isolates (with the model *E. coli* strain MG1655 as comparison). **(B)** Tolerance in *K. pneumoniae* is associated with spheroplast formation. Aliquots from the experiment depicted in panel **(A)** were transferred to agarose slides at the indicated time-points and imaged. The shown panels are representatives of multiple imaged fields **(C)** Recovery after addition of purified KPC-2 carbapenemase demonstrates spheroplast viability. Scale bar = 10 μm.

Class aPBPs like aPBP1b have an established role in rebuilding the cell wall after damage [14,23], and thus serve as internal validation of our screening conditions, establishing confidence in our TnSeq approach. Next, we validated our most prominent hits (i.e., those with the highest insertion rate pre- vs post-exposure). We used an ordered transposon insertion library in the intermediate tolerance strain KPNIH1 for a first-pass validation. We chose the KPNIH1 strain due to easy access to a large number of mutants through this library, which was generated by the Manoil lab [22]. This resulted in confirmation of several genes whose inactivation either promoted or reduced tolerance (Fig 3C). The reduced tolerance fraction contained cell envelope functions, albeit some without statistical significance, due to the high variance in this pilot screening assay (Fig 3C, green bars). An involvement of cell envelope functions is expected due to the need

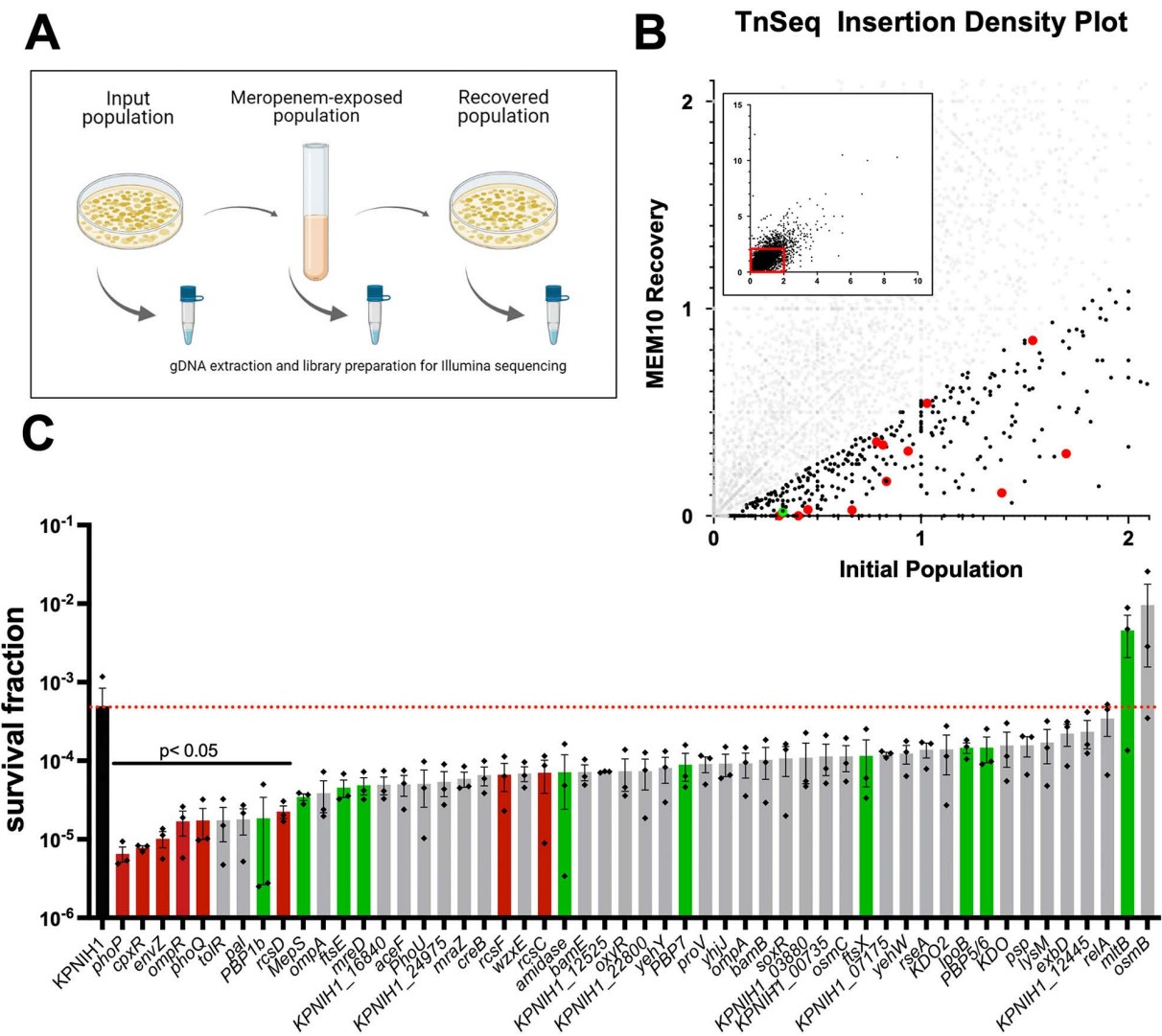

**Fig 3. TnSeq identifies genetic determinants of meropenem tolerance. (A)** Outline of TnSeq experimental procedure. Created in Biorender (Doerr, T. 2026. https://app.biorender.com/citation/69494ab2136772694a43eca2) **(B)** Identification of candidate tolerance factors in TS1 by plotting insertion density of a recovered library vs. the input library. Shown is a scatterplot of Tn insertion frequencies before (x-axis) vs. after (y-axis) meropenem exposure (inset indicates all genes, main panel represents a zoomed version of the red box in the inset). *Bona fide* tolerance genes are those that have a lower insertion frequency after exposure vs. pre-exposure. Red and green dots represent examples of tolerance genes prioritized for further validation **(C)** Validation of TnSeq hits. Mutants were obtained from the KPNIH1 mutant library and exposed to meropenem (10 µg/mL) for 6 hours, followed by dilution and spot-plating. Shown is survival fraction (CFU/mL at T6/CFU/mL at T0). Raw data of 3 independent biological replicates are shown +/- standard deviation. Red dotted line indicates WT average survival level. Red bars indicate cell envelope stress response systems; green bars represent genes involved in cell wall homeostasis. Statistical significance was established via Kruskal-Wallace test comparing each mutant to WT.

to re-establish a cell wall upon spheroplast recovery. We also noticed that disruptions in several cell envelope stress response systems (*rcs*, *pho*, *cpx*, and *ompR-envZ*), drastically and statistically significantly reduced survival during meropenem stress (Fig 3C, red bars), while two gene disruptions (in the gene for the cell wall lytic enzyme MltB, and in the poorly characterized outer membrane lipoprotein OsmB), increased survival. Taken together, these results demonstrate that the *K. pneumoniae* TnSeq experiment was effective at identifying specific genes with roles in meropenem tolerance, lending themselves to further study as outlined below.

## The lytic transglycosylase MltB endogenously reduces meropenem tolerance

Our TnSeq screen identified both genes that promote meropenem tolerance as well as genes whose products are actually detrimental in the presence of antibiotic. Inactivation of the gene coding for the lytic transglycosylase (LTG) enzyme MltB, for example, answered the screen as imparting a fitness advantage in the presence of meropenem, suggesting that MltB plays a role in meropenem-mediated killing. The specific phenotype for MltB was unexpected, since bacteria generally have many functionally redundant lytic transglycosylase paralogs throughout their genomes, and phenotypes associated with a single deletion are thus rare [24]. To dissect the role of LTGs in meropenem tolerance further, we tested a panel of transposon disruption mutants in known and predicted LTG genes for meropenem tolerance in the lower tolerance strain KPNIH1. Among eleven mutants tested, only *mltB::tn* and (to a lesser degree) *sltY::tn* exhibited significantly increased survival in the presence of meropenem, with a drastic 1000-fold increase in survival (compared to WT) after 24 hours for *mltB* and 100-fold for *sltY* (Fig 4A). Time-dependent killing experiments confirmed a strong role for MltB as an antagonist of antibiotic tolerance, as the *mltB::tn* mutant exhibited minimal (5-fold) killing in the presence of meropenem over 24 hours (Fig 4B); this phenotype could be complemented by expressing inducible *mltB* from a neutral chromosomal locus (Fig B in S1 Text).

The *mltB::tn* and *sltY:tn* mutants were further characterized by examining cell morphology upon spheroplast formation. Timelapse microscopy of cells exposed to meropenem revealed that the *mltB::tn* and *sltY:tn* disruption mutants were clearly damaged by the antibiotic (forming spheroplasts). However, *mltB::tn* retained morphology at the cell poles for longer than the parent strain (Fig 4C, red arrow), and qualitatively exhibited an increase in the relative abundance of intact spheroplasts (Fig 4C). These data suggest that *mltB* and (to a lesser extent) *sltY* may reduce tolerance by accelerating the degradation of peptidoglycan when synthesis is halted by meropenem, and thus point to a dominant role of these LTGs in cell wall turnover in *K. pneumoniae*. Perhaps activation of autolysins could be a useful future strategy to potentiate β-lactam antibiotics against tolerant pathogens.

## Multiple cell envelope stress response systems collectively contribute to β-lactam tolerance

Several well-characterized cell envelope stress response systems (CESRs) appeared in our screen as top candidates that promote meropenem tolerance: the Rcs phosphorelay system, the PhoPQ and CpxAR two-component systems, and the osmotic and acid stress response TCS, OmpR/EnvZ. Of note, our previous work revealed the PhoPQ system as a key meropenem tolerance determinant in the fellow Enterobacterales, *Enterobacter cloacae*, suggesting that PhoPQ-mediated tolerance may be a widely conserved [25]. To dissect the individual and combined contributions of these systems to survival in meropenem, we created deletions of all four systems in all combinations, including one quadruple system deletion mutant lacking all four stress responses, in our hypertolerant TS1 strain. These mutants were then exposed to meropenem in time-dependent killing assays. These experiments revealed a 10–10,000-fold decrease in viability after 24 hours of exposure to meropenem in any individual envelope stress response mutant compared to the wild type *K. pneumoniae* parent strain (Fig 5A and 5B), which could be complemented by expressing each respective system from a neutral chromosomal locus (Fig C in S1 Text). We observed large differences in both magnitude and consequences for lysis in the different mutants. The Δ*ompR/envZ* mutant, for example, exhibited the most drastic reduction in viability of all the single mutants (10,000-fold vs. 10- to 100-fold for the others), pointing to a particularly important role of the OmpR/EnvZ response in tolerance to meropenem. We also noticed that while most single mutants exhibited both a drop in CFU/mL and in $OD_{600}$ (indicative of cell lysis), the Δ*rcsF* mutant was only defective in viability, with only a slight reduction in OD during meropenem treatment (Fig 5A), suggesting a role for RcsF in spheroplast recovery, rather than structural maintenance.

We also investigated cell morphologies of each mutant before and after exposure to meropenem for 24 hours of treatment, and upon recovery in strains capable of forming spheroplasts. Wild type cells and all single deletion mutants appeared similar in shape and morphology at the start of meropenem exposure experiments (Fig 5C). However, the Δ4

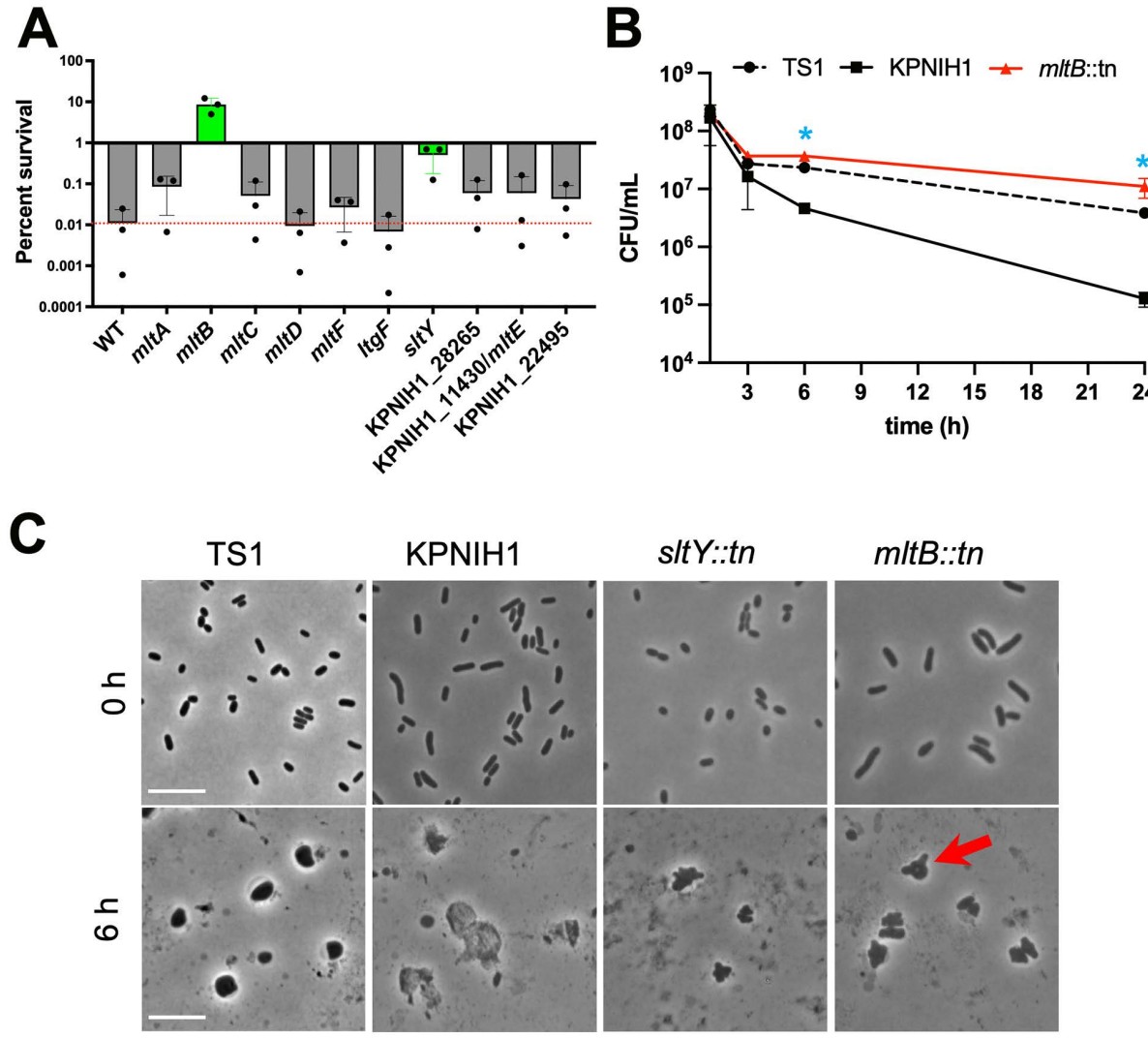

**Fig 4. MltB and SltY reduce tolerance to meropenem and prevent cells from retaining polar morphology. (A)** Survival of mutants in lytic trans-glycosylases. Mutants were obtained from the KPNIH1 mutant library and exposed to meropenem (10 µg/mL) for 6 hours, followed by dilution and spot-plating. **(B)** Time-dependent killing reveals increased tolerance in an *mltB* mutant. Shown are mean+/- SD of 6 biological replicates for *mltB::tn* (3 replicates for both WT) * = p < 0.05, Mann Whitney test. **(C)** Cell wall degradation in LTG mutants. Bacteria were treated as described for A) and then imaged at the indicated timepoints. Red arrow points to an example of a retained pole in *mltB::tn.* The hypertolerant TS1 strain is shown as comparison. Example images from 2 independent experiments with similar results are shown. Scale bars, 10 µm.

cells exhibited slightly aberrant morphology (slightly larger cells with subtle deformations) and spontaneous lysis, suggested by the presence of cell debris. After 24 hours of meropenem exposure, the WT formed robust spheroplasts, as previously observed. In contrast, spheroplast numbers were dramatically reduced in the Δ*ompR/envZ*, Δ*cpx* and Δ*phoPQ* mutants (virtually absent from the *ompR/envZ* mutant). The *rcsF* mutant formed robust spheroplasts, consistent with the relatively minor reduction in OD$_{600}$ post-treatment (Fig 5C). Thus, the Rcs system may play a role in spheroplast recovery, rather than cell envelope maintenance, consistent with a role in spheroplast recovery in lysozyme-treated *E. coli* [26]. Interestingly, the Δ*ompR/envZ* and Δ4 mutants were highly defective in spheroplast formation or maintenance, as we failed to find any intact spheroplasts even upon 10-fold concentration of meropenem-treated culture followed by imaging,

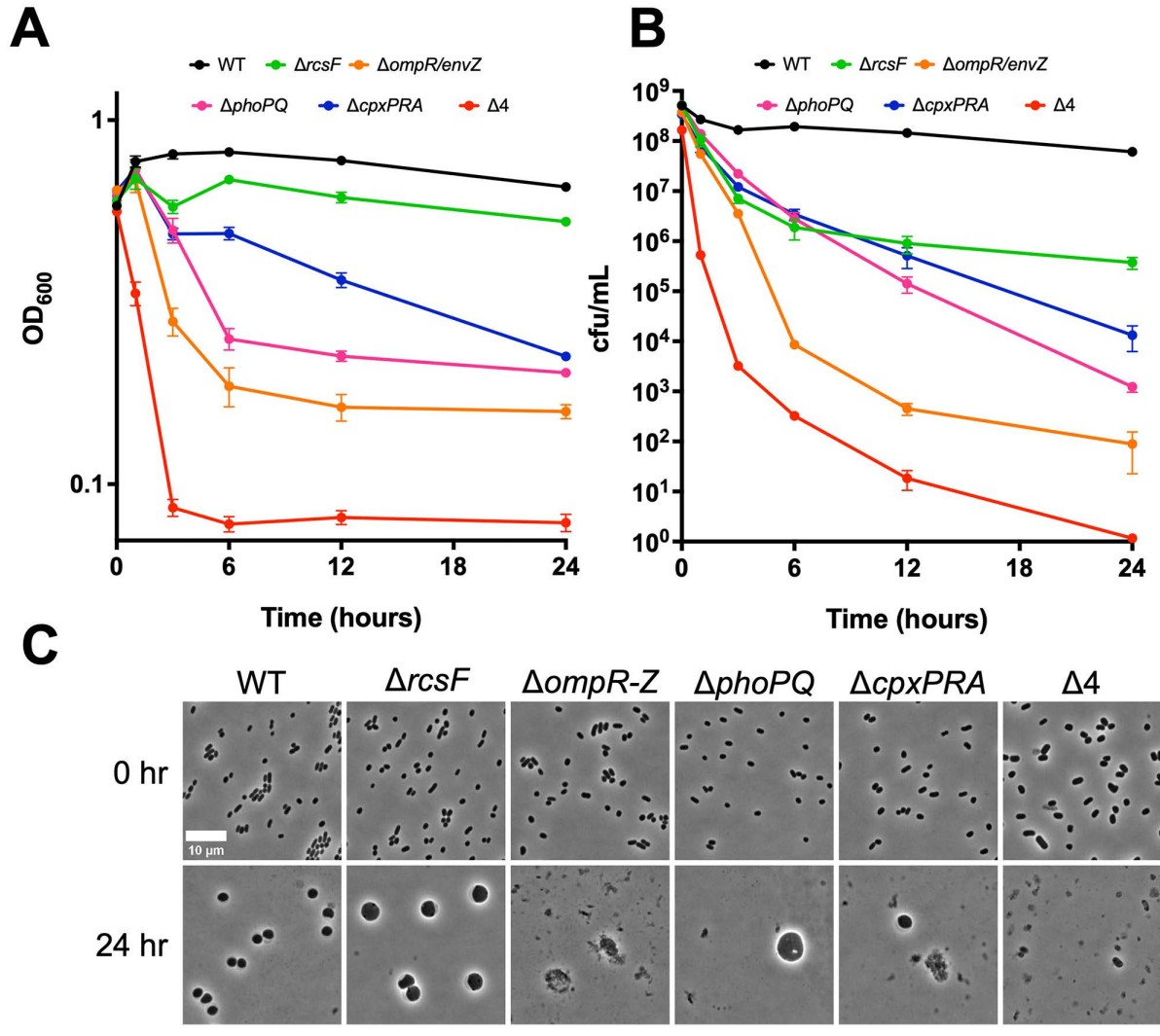

**Fig 5. Cell envelope stress response systems are essential for full meropenem tolerance.** TS1 and its CESR mutant derivatives were diluted 10-fold into 5 mL of BHI medium containing 10 µg/mL meropenem. At the indicated timepoints, samples were withdrawn and $OD_{600}$ **(A)** and CFU/mL **(B)** were measured. Data represent means +/- standard deviation of at least 4 biological replicates. At 24 hours, differences between each mutant and WT, and between Δ4 and each single mutant except Δ*ompR*/*envZ* were statistically significant (Mann Whitney U test, p = 0.002). **(C)** Aliquots were removed from the experiment depicted in (A-B) and imaged. Scale bar = 10 µm.

consistent with the drastic reduction in viability these strains experience in the presence of meropenem (Fig 5B). We also constructed intermediate strains, including a suite of Δ3s with single two-component systems left intact (Fig D in S1 Text). All Δ3s collapsed in viability almost to the same degree as the Δ4, strongly suggesting an additive or synergistic effect between the different systems, as no two-component system by itself could sustain viability. Of note, among the Δ2 mutants, the combination of Δ*ompR*/*envZ* and Δ*rcsF* was most detrimental, and the resulting mutant was, like the Δ4 mutant, nearly eradicated after 24 hours of exposure. Collectively, these results point to a functionally hierarchical organization of responses necessary for meropenem tolerance: OmpR/EnvZ as the most important response, followed by Pho and Cpx to maintain spheroplast stability, and Rcs with a more prominent role in recovery compared to maintenance of structural integrity.

## The Rcs phosphorelay is induced by, and required for, meropenem tolerance

The Rcs phosphorelay as an apparent contributor to tolerance caught our attention. In *E. coli*, Rcs is induced by β-lactams, and required for survival in low β-lactam concentrations [27]. Additionally, Rcs is required for recovery of *E. coli* from a cell wall-deficient state (generated by lysozyme treatment in osmostabilized medium, a condition that morphologically mimics meropenem-treated spheroplasts) [26]. In apparent contrast, a recent study has shown that Rcs is neither induced by, nor required for β-lactam persisters [28]; however, this is likely because persisters are dormant and thus not expected to be damaged by the antibiotic (obviating the need for a damage response) [3]. To clarify the role of Rcs in *K. pneumoniae*, we engineered a TS1 strain carrying an $P_{rprA}$-*lacZ* transcriptional fusion (a well-established readout of Rcs signaling [29]) in a neutral chromosomal locus (*lacZ*) and measured induction by meropenem. A first qualitative assay using zone of inhibition on growth medium containing the chromogenic LacZ substrate X-gal yielded a blue ring around the zone of inhibition when a meropenem disk was placed on the agar surface, but not when the translation inhibitor kanamycin was used (Fig E in S1 Text). We then corroborated these results in liquid medium, where a quantitative β-galactosidase activity assay (Miller assay) revealed a 5-fold induction of the construct by meropenem that was largely dependent on the presence of RcsB (Fig 6A). Thus, meropenem induces the Rcs response. We also validated the role of Rcs in meropenem tolerance by creating additional mutants in Rcs components, namely RcsC and RcsD. All mutants displayed a strong (10,000-fold) reduction in tolerance, which could be fully complemented by overexpression of the response regulator RcsB (Fig 6B). The ability of RcsB overexpression to induce the Rcs regulon was verified by measuring induction of the chromosomal $P_{rprA}$-*lacZ* construct (Fig E in S1 Text). Thus, the Rcs phosphorelay is both responsive to meropenem exposure, and required for meropenem tolerance in *K. pneumoniae.*

## Fluctuation test reveals memory in tolerant subpopulation

The stress responses that contribute to meropenem tolerance may rely on deterministic induction by the antibiotic for their protective impact, or they may be stochastically and heterogeneously active within the population before exposure to a stress as a bet-hedging strategy. To gain some insight into this distinction, we leveraged the classical Luria-Delbrück experiment or the "Fluctuation Test" that has been recently expanded to investigate reversible switching between cellular states within an isogenic cell population. This fluctuation test has been used to gain insights into the non-genetic mechanisms by which microbial and cancer cells survive lethal exposure to targeted drug therapy [30–32]. For example, the application of the fluctuation test in *Escherichia coli* [33], *Bacillus megaterium* [34] and *Mycobacterium tuberculosis* [35] has shown individual cells to reversibly switch between an antibiotic-sensitive and an antibiotic-tolerant state, with each state being maintained transiently for several generations. This reversible switching happens continuously in the absence of the drug as a result of stochastic fluctuations in underlying signaling and gene regulatory networks and primes a rare subpopulation of cells as a bet-hedging mechanism to survive lethal antibiotic exposure.

To conduct a fluctuation test, we first isolated single TS1 cells in 96-well plates using serial dilution. Each cell was grown into a clonal population and exposed to meropenem at an $OD_{600}$ of 0.3 (Fig 7). WT OD dropped sharply at first, followed by slower increase in $OD_{600}$ in the surviving subpopulation (spheroplasts). The clone-to-clone fluctuations in cell viability after antibiotic exposure were quantified by using the coefficient of variation (standard deviation divided by the mean) of $OD_{600}$ (spheroplast survivor fraction) across clones at 1.66 hours after drug exposure. This time point was specifically chosen as it captures the $OD_{600}$ drop-off in response to meropenem (Fig 7, dashed vertical blue line) and avoids the rebound in $OD_{600}$ seen in WT cells as a result of changes in cell shape. We then used mathematical modeling to predict how the kinetics of switching between drug-sensitive and drug-tolerant cell states impact clone-to-clone fluctuations in the number of cells surviving lethal stress, with slower switching (i.e., higher transient memory of a state being retained across generation) resulting in a higher degree of interclonal fluctuations [32]. Our results show clone-to-clone fluctuations in cell integrity to be 0.153 ± 0.036 for WT where ± denotes the 95% confidence intervals in the coefficient of variation of $OD_{600}$ as obtained by bootstrapping (Fig 7). These observed fluctuations were significantly higher than the technical noise

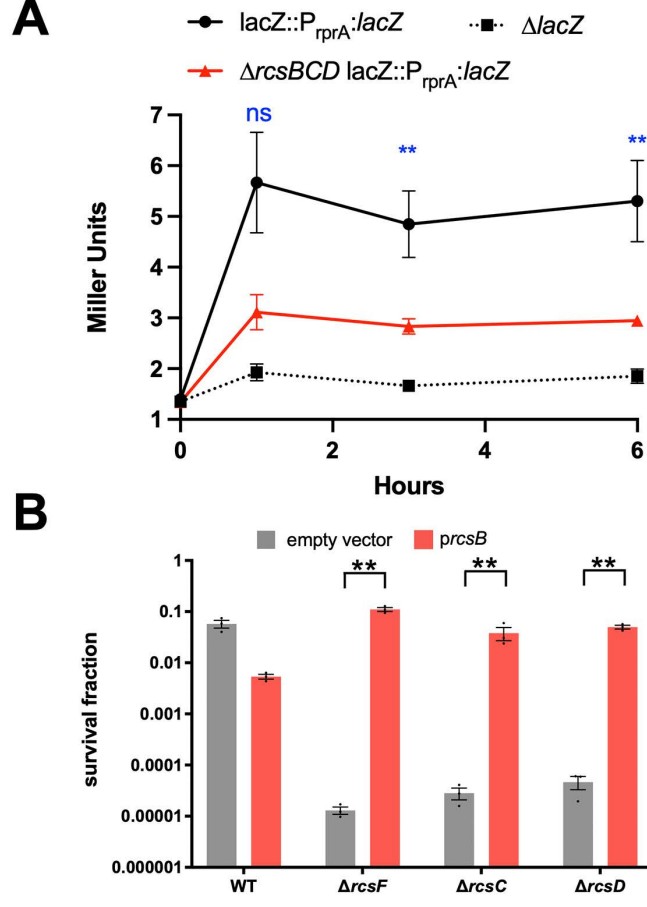

**Fig 6. The Rcs phosphorelay is induced by meropenem and required for tolerance. (A)** WT and Δ*rcsBCD* strains carrying a chromosomal rprA-*lacZ* transcriptional fusion were exposed to meropenem, followed by quantification of β-galactosidase activity via Miller assay. A strain deleted in *lacZ* was used as background control. ** = p < 0.05, ns = not significant (t-test) **(B)** The indicated strains were exposed to meropenem (10 µg/mL). Survival fraction is CFU/mL after 24 hours normalized to starting cell density. Shown are averages of 3 independent experiments + /- standard error. Statistical significance was tested via Welsh's t-test on log-transformed data (** = p < 0.05). Each single mutant was also statistically significant from WT (p < 0.05, Welsh's t-test on log-transformed data, omitted from graph for clarity).

of 0.086 ± 0.028 (control fluctuations). Technical noise was estimated by performing a similar experiment, but with fluctuations quantified across wells starting with a random population of 10,000 cells rather than a single cell-derived population. These results were statistically significant (testing for equality of CVs [36]; p = 0.0031 for WT 1 cell vs. $10^4$ cells). Similar to observations in other bacteria, this fluctuation data is consistent with a model of a multi-generation drug-tolerant state in *Klebsiella pneumoniae* that preexists meropenem exposure.

We hypothesized that the stochasticity originated from variance in background expression of cell envelope stress responses. To test this, we repeated the fluctuation assay in the Δ4 genotype where the observed fluctuations were much smaller compared to WT with the coefficient of variation being 0.044 ± 0.009 (p = 9.7 x $10^{-6}$ one cell vs. $10^4$ cells), but still significant compared to control fluctuations performed for this strain (Fig 7). The lower value of the observed clonal fluctuations in Δ4 points to faster-switching kinetics, and hence a shorter transient memory of the drug-tolerant state, suggesting that at least part of the heterogeneity in tolerance levels is caused by stochastic baseline expression levels of cell envelope stress response systems.

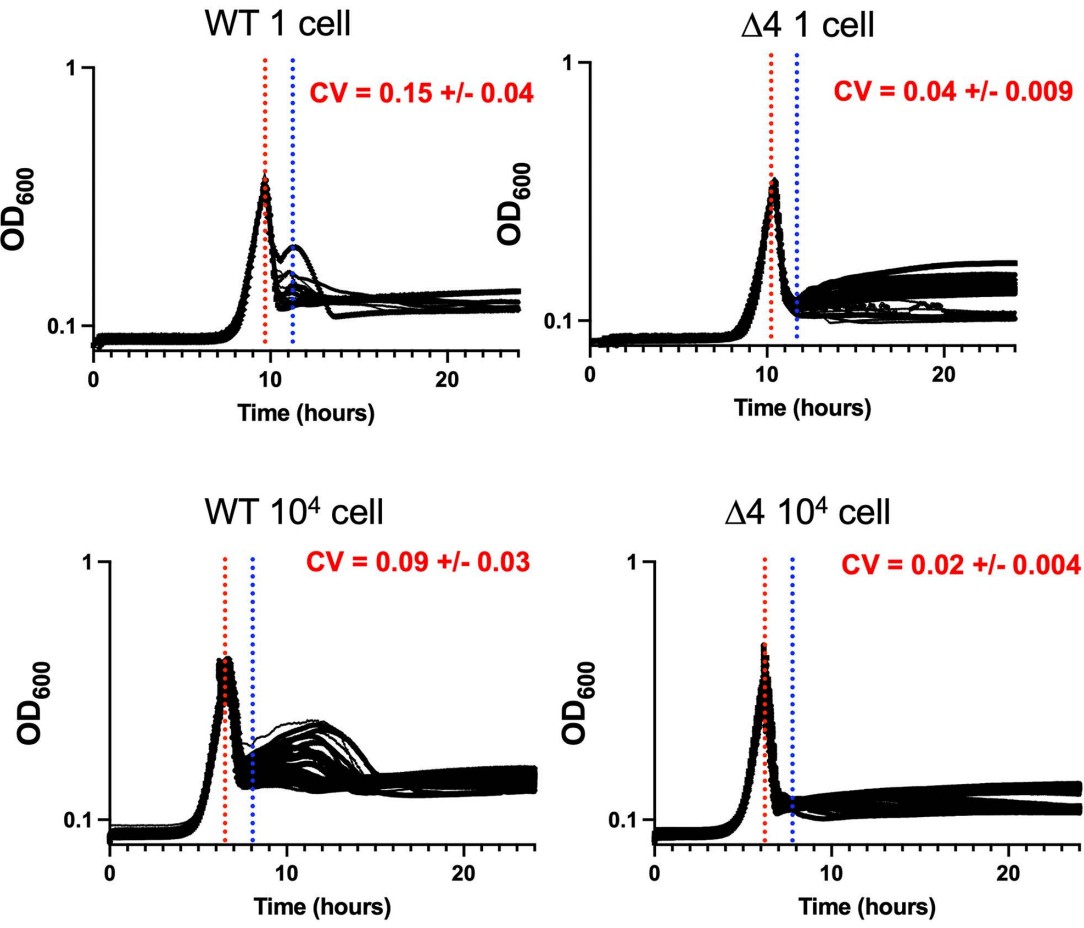

**Fig 7. Fluctuation test indicates memory in spheroplast population.** Bacteria were diluted to a starting density of 1 cell/well or 10,000 cells/well (control), and grown in a microtiter plate. At $OD_{600} = 0.3$, meropenem (10 μg/mL) was added (red dotted line). CV (coefficient of variation) calculations were performed at peak killing (blue dotted line).

## Discussion

Infections with Enterobacterales continue to be an active and increasing threat to public health due to their capacity to cause infection, and to gain and disseminate antibiotic resistance genes, with particular risk for vulnerable populations. In this work, we have found that antibiotic tolerance, the ability to sustain viability in the presence of bactericidal antibiotics for extended time periods, is widespread among clinical isolates of Gram-negative pathogens, particularly *Klebsiella pneumoniae*.

To uncover novel factors that modulate β-lactam tolerance in *K. pneumoniae* we performed a genome-wide scan using a TnSeq approach. Because this approach provides a readout of each gene's fitness during a challenge, we were able to identify genes that contribute to meropenem tolerance, and genes that reduce tolerance in *K. pneumoniae*. The screen overall was able to capture several regulatory systems and individual factors that modulate tolerance along with the internal control PBP1b (a key cell wall synthase required to rebuild the cell wall after catastrophic damage [23]), raising confidence in our approach. This screen also pointed to several cell envelope stress responses that are partly responsible for this strain's ability to undergo the entire process of spheroplast formation, maintenance, and recovery: the Rcs, Pho, Cpx, and OmpR/EnvZ systems. Of note, the ability of cell envelope stress responses to confer antibiotic tolerance via repair of antibiotic-induced damage appears to be widespread among diverse bacteria [11,13,37,38].

PLOS Pathogens

Interestingly, the OmpR/EnvZ system appears to exert a particularly strong effect on tolerance. This system is induced by changes in pH and osmolarity, and in *E. coli* controls the switch from the larger diameter porin OmpF to the smaller diameter porin OmpC [39]. Formally, it is possible that the porin switch may delay diffusion of meropenem into the cell; however, this is unlikely to account for the tolerance phenotype we observe here, given that WT and Δ*ompR/envZ* MICs are within essential agreement (S2 Table). Future work will be directed at delineating the contribution of this system to spheroplast maintenance. Conversely, the Pho and Cpx single system deletion mutants appear to have similar net contributions to meropenem tolerance both in rate of killing and final viability, while still forming detectable spheroplasts capable of recovery. The versatile PhoPQ system senses $Mg^{2+}$ deficiency, presence of antimicrobial peptides, pH and other stresses [40]. Most relevant to the phenotype assayed here, we have previously shown that PhoPQ is required for meropenem tolerance in the related bacterium *Enterobacter cloacae*, via PhoPQ-controlled outer membrane modifications [25]. The Cpx system is a general cell envelope health monitor [41], which controls, among other things, L,D-transpeptidases that help stabilize the outer membrane [42,43], and respiratory chain functions [44,45], which contribute to ROS production in spheroplasts [11]. Lastly, the Rcs system contributes to meropenem tolerance in a unique way. Time-dependent killing revealed that viability dropped substantially in the Δ*rcsF* mutant, but this strain's capacity to form spheroplasts and maintain optical density was comparable to the WT parent strain. However, Δ*rcsF* mutant spheroplasts were markedly impaired in their ability to reform successful dividing rods when meropenem is removed. Among other genes, Rcs positively regulates the cell division component *ftsZ* [46], perhaps pointing to a model where upon spheroplast recovery, cell division can commence sooner in WT than in Δ*rcs* cells due to the increased availability of critical divisome components. Taken together, the phenotypes of each individual mutant indicate that aspects of β-lactam tolerance are mediated by distinct response systems, likely with some temporal distinction (Fig 8). These responses (or at least the general strategy of cell envelope damage repair via stress response systems) appear to be a well-conserved mechanism of β-lactam tolerance, as stress response systems (including some of the systems revealed here) have been identified as contributors to survival in the presence of β-lactams in *E. coli* [47], *Acinetobacter baumanii* [48], and *Burkholderia pseudomallei* [20].

Because TnSeq has the benefit of also revealing gene disruptions that increase fitness, we simultaneously probed for endogenous factors that reduce meropenem tolerance. *K. pneumoniae* strain TS1's lytic transglycosylase *mltB* answered the screen as a strong candidate that actually improves survival when disrupted, which conversely suggests that MltB contributes to cell killing in the presence of meropenem. Lytic transglycosylases cleave glycosidic bonds of peptidoglycan strands [24], and have specific and important functions in opportunistic pathogens. For example, in *Acinetobacter baumannii*, *mltB* is a critical factor for pathogenesis by linking complex envelope homeostasis functions with virulence [49]. In *Pseudomonas aeruginosa*, eliminating this enzyme (also in combination with other lytic transglycosylases) increases resistance to cell-wall targeting antibiotics via induction of an endogenous β-lactamase [50,51]. Disruption of *sltY* and *mltB* in *K. pneumoniae* strain KPNIH1 resulted in 10- to 1000-fold increased tolerance to meropenem. Cell morphology around the poles in these mutants was maintained after exposure to meropenem, possibly due to this strain's inability to degrade the old peptidoglycan present when PBPs are inhibited. This maintenance of the cell wall could be what increases survival by preserving the peptidoglycan and envelope-spanning systems, leading to preservation of cell integrity and potentially serving as a scaffold for new cell wall synthesis when meropenem is removed. It is unclear why *mltB* and *sltY* specifically emerge as the only lytic transglycosylases that have this effect on the cell when bacteria generally have multiple redundant lytic transglycosylase paralogs in their genomes; strong phenotypes for single deletion mutants are thus highly unusual. In *E. coli,* deletion of especially *slt70* has a somewhat paradoxical effect, where the mutation confers hypersensitivity to the antibiotic mecillinam (i.e., the opposite phenotype to the one observed here), but confers resistance when paired with activation of the divisome through induction of *ftsZ*. The interpretation in this case is that since mecillinam targets the Rod system for cell elongation, resistance requires a combination of reducing futile cycling (by deleting *slt70*) and promoting division in the presence of antibiotic (by activation of the FtsZ ring) so that morphologically aberrant cells

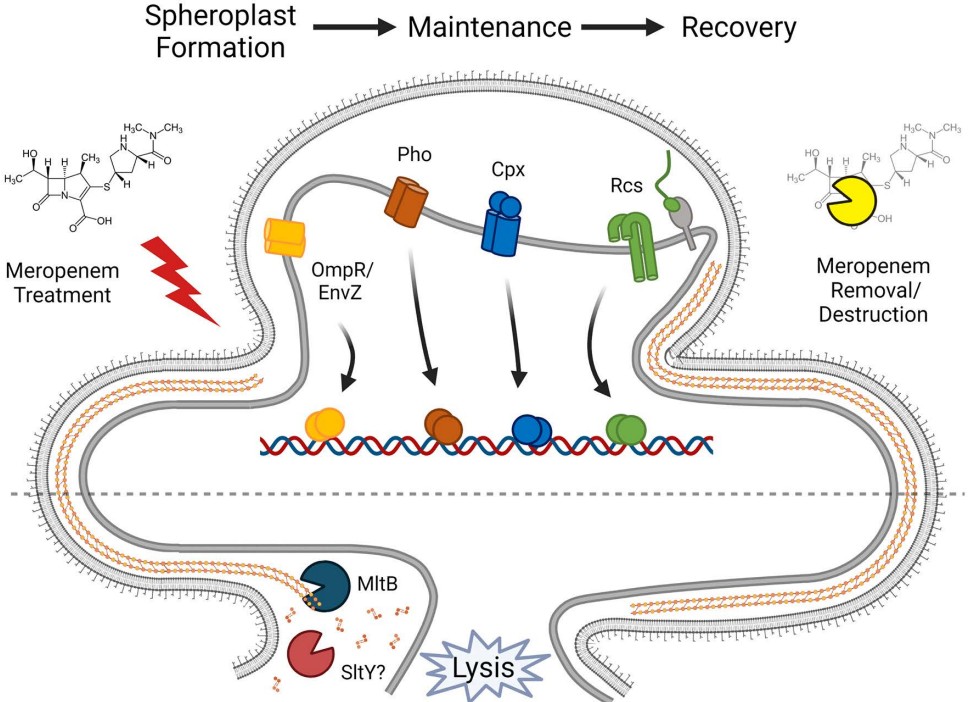

**Fig 8. Summary of meropenem tolerance as a net result of cell wall degradation and cell envelope stress response induction.** Meropenem treatment induces multiple cell envelope stress responses, whose regulons contribute to survival. Lytic transglycosylases MltB and SltY contribute significantly to sacculus degradation, which reduces efficiency of spheroplast recovery. Partially created in Biorender (Doerr, T. 2026. https://app.biorender.com/citation/6945955a37d46757be2b127c).

can still divide [18]. It is possible that the tolerance increase we observe in the *mltB/sltY* mutants likewise reflects reduced futile cycling, though importantly, spheroplasts do not divide in the presence of antibiotic. We noted that both LTG mutations also increased meropenem MIC significantly (S2 Table). It is thus possible that increased degradation of meropenem, for example by overproduction of a chromosomal β-lactamase with low-level carbapenemase activity, also at least partially contributes to enhanced survival of these mutants (as a caveat, *K. pneumoniae's* chromosomal enzyme SHV-1 is not known to hydrolyze carbapenems). However, these mutants still form spheroplasts, indicating significant cell wall damage, ergo meropenem is still present at supra-MIC concentrations. Putative carbapenemase activity can thus not account entirely for the high survival phenotype.

Overall, this study presents a deeper insight into β-lactam tolerance and can inform further investigations to understand other aspects of how these dangerous pathogens respond to antibiotics and ultimately develop resistance. Attaining an understanding of the detailed workings of this complex phenotype can inform approaches toward finding potential druggable targets to reduce or abolish an organism's capacity for β-lactam tolerance, thus allowing development of novel therapeutics and treatment strategies for tolerant pathogens.

## Methods

*Culture conditions and killing experiments* Unless otherwise stated, all strains were grown in shaking overnight cultures at 37ºC, and tolerance assays were carried out in Brain Heart Infusion (BHI) broth at 37ºC. Tolerance assays were conducted by inoculating overnight culture at a 1:10 ratio into fresh BHI medium containing 10 μg/mL meropenem (TCI, Tokyo, Japan). Cultures were then incubated at 37˚C without shaking. Optical density was measured in

a VWR V-1200 spectrophotometer and viable cells were quantified by serial dilution in BHI medium and spot-titered onto BHI agar. The carbapenemase KPC-2 was purified (see next section for detail) and added to culture samples at a final concentration of approximately 1 µg/mL prior to serial dilution to deactivate meropenem, thus eliminating meropenem carryover effect. For complementation experiments, 40 ng/mL anhydrotetracycline was added for induction of expression.

## MIC Determination

MICs for all strain backgrounds (except the isolate panel, see below) were determined by diluting overnight cultures $10^7$-fold in BHI medium for direct comparison with killing experiments. Diluted cultures were inoculated into BHI containing successive 2-fold concentrations of meropenem in a 96 well microtiter plate at a final volume of 200 µL per well, and incubated at 37°C overnight, leaving one well as an uninoculated control, and one well with cells but without meropenem to ensure growth of the strain. MICs were recorded as the lowest meropenem concentration free from turbidity. For the clinical isolate panel, MICs were measured using standard CLSI guidelines (cation-adjusted MHB).

## Purification of KPC-2 enzyme

The gene for KPC-2 was amplified from the genome of *Enterobacter cloacae* strain 41952, a clinical isolate from WCM, and cloned into the pET21 expression vector. The His-tagged KPC-2 enzyme was overexpressed in BL21 pLysS. A 500 ml culture of LB containing 100 µg/mL ampicillin was induced at OD = 0.5 with 1 mM IPTG and incubated shaking at 30°C overnight. The following day, cultures were centrifuged, and pellets were decanted and stored at -80°C until lysis and purification. Pellets were resuspended in resuspension buffer (20 mM Tris, 150 mM NaCl, 5 mM imidazole) and sonicated using the QSonica with the largest tip attachment for 40 minutes total, alternating between 5 second pulses and 5 second rests on ice. The lysate was then centrifuged for 45 minutes at 16,000 rpm and the supernatant discarded. In preliminary expression experiments, we found that KPC was mostly in the insoluble fraction. The insoluble pellet was resuspended in binding buffer (20 mM Tris, 150 mM NaCl, 5 mM imidazole, 3 M Urea), and incubated rotating overnight at 4°C to denature and solubilize the KPC-2 protein. The next day, the denatured lysate was centrifuged again at 16,000 rpm for 45 minutes, and the resulting supernatant passed through a cobalt-charged NTA agarose resin column at room temperature. Six column volumes of wash buffer (20 mM Tris, 150 mM NaCl, 30 mM imidazole, 3 M Urea) were then passed through the column, and the protein eluted in 1.8 mL fractions with increasing concentrations of imidazole in elution buffers (20 mM Tris, 150 mM NaCl, 50–600 mM imidazole, 3 M Urea). Fractions with the highest protein concentrations were pooled, and the total volume was dialyzed into a final storage buffer (20 mM Tris, 150 mM NaCl, 30% glycerol) in three steps, each with decreasing concentrations of imidazole and increasing concentrations of glycerol. Purity was determined by SDS-PAGE, and concentration was estimated with Bio-Rad Bradford Assay kit.

## Imaging

All images were taken on a Leica Mdi8 microscope (Leica Microsystems, GmbH, Wetzlar, Germany) with a PECON TempController 2000–1 (Erbach, Germany), heated stage at 37°C for growth experiments, or room temperature for static images. For timelapse microscopy of spheroplast formation, cells were placed on a 0.8% agarose pad containing BHI and 10 µg/mL meropenem within a gene frame (ThermoFisher, Waltham, MA). For recovery timelapses, KPC-2 enzyme was added at 5 µg/mL to eliminate the meropenem. Frames were taken using autofocus control every five minutes for up to 12 hours. Timelapse video stacks were processed in FIJI (NIH) using the Linear Stack Alignment with SIFT under the Registration plugin (using all default settings with the exception of Expected Transformation, which was changed to Translation) to center cells in the cropped frame.

## Transposon mutagenesis screen and data analysis

*K. pneumoniae* strain TS1 was mutagenized with a Mariner transposon delivered via the *E. coli* donor strain MFD λpir to generate a complex library of kanamycin-resistant transposon mutants. This initial library was frozen as a glycerol stock, and an overnight culture from it was grown in BHI containing kanamycin at 50 μg/mL to form an input library as the comparison for TnSeq analysis. Overnight cultures of this input library were diluted 10-fold into 5 mL BHI containing meropenem (10 ug/mL) and incubated for six hours at 37˚C without shaking, followed by plating on BHI agar plates after treatment with 5 μg/mLKPC-2 enzyme (untreated controls were also generated mirroring the six-hour exposure and recovery populations). Genomic DNA (gDNA) was extracted from all stages of the meropenem challenge and recovery and prepared for Illumina sequencing. Briefly, gDNA extracted from each library was sonicated to generate fragments mostly 200–800 base pairs long which were blunted using NEB Quick Blunting Mix (#E1201L). After blunting, fragments were given A-tails with Taq DNA polymerase to allow ligation of adapter sequences. Ligated fragments were amplified with PCR to enrich transposon-adjacent regions, and then again were PCR amplified with primers to add barcodes, spacer sequences, and Illumina chip attachment ends. The library was run on agarose gel and all fragments 250–600 base pairs were extracted for analysis on the BioAnalyzer (Agilent, Santa Clara, CA) to assess before Illumina sequencing. All libraries were sequenced as paired 150 bp reads, though only forward reads were used in the final analysis.

Raw Illumina data were processed in Galaxy using the Cutadapt program to trim reads for length and quality, then mapped with Bowtie2 using the *K. pneumoniae* TS1 FASTA reference genome available in NCBI (Biosample accession number SAMN04014921). Mapped reads were analyzed by TnSeq Explorer [52] to calculate insertion density of each gene from each library. Insertion densities were used to calculate an insertion density ratio for each gene from one library to another, thereby comparing essentiality across points in the meropenem challenge. The comparison between library 2 (input) and library 6 (meropenem recovery) was used as the basis for ranking genes in order of essentiality, with genes most essential to meropenem tolerance appearing at the top of the list, while genes which impart a fitness increase when disrupted appearing at the bottom of the list.

## KPNIH1 validations

Candidate tolerance genes identified by TnSeq were first validated using an arrayed *K. pneumoniae* KPNIH1 library constructed by the Manoil lab [22]. Transposon mutants were grown in 200 μL BHI medium overnight cultures in a 96 well plate. The next day, the overnight culture was diluted 1:10 into BHI containing a final concentration of 10 μg/mL meropenem. At the indicated time points, samples were taken for CFU determination, while simultaneously measuring optical densities in a Spectramax iX3 (Molecular Devices, San Jose, CA) plate reader, and then incubated at 37˚C without shaking for six hours. Final optical density and CFU/mL were recorded and the ratio between final and initial values were calculated to determine the effect of the gene disruption when challenged with meropenem.

## Strain constructions with pTOX

Strains and oligos are summarized in S5 and S6 Tables. Knock out and reporter strains in *K. pneumoniae* TS1 were constructed using the pTOX5 allelic exchange vector as described in [53]. Briefly, 800–1000 base pair regions flanking regions were amplified using PCR or synthesized de novo and assembled via isothermal assembly into the pTOX SmaI site. Assembled constructs were transformed into DH5α λpir competent cells, plated on LB containing 100 μg/mL chloramphenicol and 1% glucose (LB/CHL/gluc) (to repress the rhamnose inducible toxin gene) and screened for insertion size with primers TC_82 and TC_83, followed by sequence verification through Sanger sequencing. Correct constructs were transformed into conjugal MFD λpir and mated into *K. pneumoniae* TS1 on LB containing 1% glucose and 1.2 mM DAP (LB/gluc/DAP) for 6 hours at 37°C. Transconjugants were isolated back on LB/CHL/gluc and a single colony was grown out in LB containing 1% glucose until the OD reached 0.1-0.2, at which point cells were washed with M9 medium

containing 0.2% casamino acids and 1% rhamnose, and plated on solid M9/rham/casamino acids to induce the counterselective toxin and isolate allelic exchange constructs containing the desired modifications. Envelope stress response system deletions were screened using primers flanking the loci of interest and validated using internal primers. Other constructs were validated using primers specific to the modification, or primers flanking the *lacZ* locus where reporters were inserted.

For complementation of phenotypes, genes were cloned into a pTn7 plasmid (tn7-gentamycin) and inserted into the chromosomal Tn7 insertion site (near the *glmS* gene) using triparental mating with the helper plasmid pTNS2. The Tn7 plasmid was modified to contain an anydrotetracycline-inducible (Tet on) gene fragment (DL-GB-PP-1, see S6 Table). MltB and CESRs were amplified from the chromosome of Kp KPNIH1 (*mltB*) and TS1 (CESR loci), respectively, using primers NN520/521 (mltB), NN518/519 (*cpxPRA*), NN512/513 (*ompR/envZ*), NN514/515 (*phoPQ*), NN516/517 (*rcsDBC*) and cloned into PacI-digested pTN7 using isothermal assembly. Successful clones were screened for using primers NN522/NN523, and verified via whole plasmid sequencing.

### Miller assay to detect RcsB function and induction

To assess function of the RcsB response regulator, a reporter construct was constructed by inserting an *rprA-lacZ*$_{MG1655}$ reporter with the *E. coli* MG1655 lacZ gene (abbreviated to *rprA-lacZ*) into the TS1 native *lacZ* locus, eliminating most of the native *lacI* and *lacZ* genes. This construct was validated first by overexpressing native RcsB on a pBAD plasmid in the presence of 0.2% arabinose on BHI medium from strain TS1 in the reporter background and qualitatively comparing X-gal signal to the construct containing an empty vector. RcsB activation upon meropenem exposure was measured by performing the Miller assay on cell pellets from meropenem-exposed cells at 0, 1, 3, and 6 hours in BHI and control cultures to determine relative units of β-galactosidase activity normalized to CFU/mL of either culture.

### Fluctuation test

Fluctuation test experiments in strain TS1 and its Δ4 derivative were done by transferring 100 μL of seed stock (overnight cultures $10^9$-fold in 15 mL BHI medium) into a 96-well plate. At this dilution, every three to four wells receives a single cell. This plate was then incubated without shaking at 37°C in a Spectramax i3x set to read optical density every five minutes. Individual wells that started with the presence of cells were grown to an optical density of 0.3, at which point they were treated with meropenem at a final concentration of 10 μg/mL, and allowed to continue incubating. Datasets were aligned such that all additions of meropenem were at a uniform OD 0.3. Control experiments were set up by inoculating each well with 10,000 colony forming units and growing to optical density 0.3, at which point all individual populations in each well were treated with 10 μg/mL meropenem as above. Co-efficient of variation (CV) in OD values was then calculated upon reaching a plateau of lysis. A higher CV at lower starting density (single cell, more generations before drug exposure) than higher starting density (10,000 cells, fewer generations before drug exposure) indicates transient memory, as increased variation in the population after extended growth indicates that variability is pre-existing, rather than induced by the antibiotic, and multigenerational.

### Supporting information

**S1 Text.** Fig A. Tolerance screen validation with benchmark isolates. Strains were exposed to meropenem (10 μg/mL) as described in methods. At the indicated time-points, 10 μL of the cell suspension was spotted on BHI agar and supplemented with KPC, followed by 24 hours of incubation at 37 ˚C. Fig B. Complementation of the Δ*mltB* phenotype. The indicated strains were grown overnight in BHI, then diluted 10-fold into fresh BHI containing meropenem (10 μg/mL) and 40 ng/mL anhydrotetracycline. Viability was determined by serial dilution and spot-plating after 24 hours of incubation. Shown are raw data points for 5 biological replicates for each strain. Statistical significance was determined by Mann

Whitney test (*p < 0.05). Fig C. Tolerance phenotypes of CESR mutants. Time-dependent killing experiments (10 µg/mL meropenem) were conducted in BHI medium. Statistical analysis (Kruskal-Wallace test, p < 0.001) revealed statistically significant differences in viability (12 hour time point) for all mutants vs. WT; Δ4 vs. any triple mutant was not statistically significant. Fig D. Complementation of the ΔCESR phenotypes. The indicated strains were grown overnight in BHI, then diluted 10-fold into fresh BHI containing meropenem (10 µg/mL) and 40 ng/mL anhydrotetracycline and incubated for 24 hours. Viability was determined by serial dilution and spot-plating. Shown are boxplot and raw data points for at least 5 biological replicates for each strain. Fig E. Meropenem induces the Rcs phosphorelay. A) The $P_{rprA}$-*lacZ* reporter strain was spread on plates containing the chromogenic LacZ substrate x-gal. A filter disk was placed in the middle containing either meropenem or kanamycin. Note blue halo around the meropenem disk, in the sub-MIC area beyond the zone of inhibition. B) RcsB was overexpressed from an arabinose-inducible promoter in the $P_{rprA}$-*lacZ* background, followed by plating on LB with X-gal.
(DOCX)

**S1 Table. Summary of Bloodstream Isolate Panel Distribution in tolerance categories.** Shown are the number of isolates for each species in each tolerance category.
(XLSX)

**S2 Table. MICs of the strains used in this study.**
(XLSX)

**S3 Table. Bloodstream isolate panel.**
(XLSX)

**S4 Table. TnSeq insertion frequencies.**
(XLSX)

**S5 Table. Strains used in this study.**
(XLSX)

**S6 Table. Oligos used in this study.**
(XLSX)

## Author contributions

**Conceptualization:** Trevor Cross, Facundo Torres, Nadia Nikulin, Abigail P McGee, Tobias Dörr.

**Data curation:** Trevor Cross, Facundo Torres, Nadia Nikulin, Abigail P McGee, Rhea Balakrishnan, Tobias Dörr.

**Formal analysis:** Trevor Cross, Facundo Torres, Nadia Nikulin, Abigail P McGee, Abhyudai Singh, Tobias Dörr.

**Funding acquisition:** Tobias Dörr.

**Investigation:** Trevor Cross, Abigail P McGee, Leena Jalees, Rhea Balakrishnan, Tolani Aliyu, Tobias Dörr.

**Methodology:** Trevor Cross, Facundo Torres, Nadia Nikulin, Abigail P McGee, Leena Jalees, Rhea Balakrishnan, Tolani Aliyu, Tobias Dörr.

**Project administration:** Tobias Dörr.

**Resources:** Lars F. Westblade, Abhyudai Singh, Tobias Dörr.

**Software:** Abhyudai Singh.

**Supervision:** Tobias Dörr.

**Validation:** Trevor Cross, Facundo Torres, Tobias Dörr.

**Visualization:** Trevor Cross, Facundo Torres, Nadia Nikulin, Leena Jalees, Rhea Balakrishnan, Abhyudai Singh, Tobias Dörr.

**Writing – original draft:** Trevor Cross, Tobias Dörr.

**Writing – review & editing:** Facundo Torres, Nadia Nikulin, Lars F. Westblade, Tobias Dörr.

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
