## [Decision Letter · Decision Letter 0]

11 Dec 2025

PPATHOGENS-D-25-02804

Prevalence and mechanisms of high-level carbapenem antibiotic tolerance in clinical isolates of Klebsiella pneumoniae

PLOS Pathogens

Dear Dr. Dorr,

Thank you for submitting your manuscript to PLOS Pathogens. After careful consideration, we feel that it has merit but does not fully meet PLOS Pathogens's publication criteria as it currently stands. Therefore, we invite you to submit a revised version of the manuscript that addresses the points raised during the review process. Of particular note, the fluctuation test needs clarification to improve the understanding of this technique. Addressing the novelty of these findings, there are previous reports not mentioned in the manuscript that have implicated the factors identified here are important for cell envelope stress. Furthermore, as the current focus of tolerance is on growth rate, the results presented here showing that stress response mechanisms preserve non-growing cells should be explained in the context of growth rate to show how these findings change our current understanding of tolerance.

We look forward to receiving your revised manuscript.

Kind regards,

Renee M. Fleeman

Guest Editor

PLOS Pathogens

Matthew Wolfgang

Section Editor

PLOS Pathogens

Sumita Bhaduri-McIntosh

Editor-in-Chief

PLOS Pathogens

orcid.org/0000-0003-2946-9497

Michael Malim

Editor-in-Chief

PLOS Pathogens

orcid.org/0000-0002-7699-2064

**Journal Requirements:**

At this stage, the following Authors/Authors require contributions: Trevor Cross, Facundo Torres, Nadia Nikulin, Abigail P McGee, Leena Jalees, Rhea Balakrishnan, Tolani Aliyu, Lars Westblade, Abhyudai Singh, and Tobias Dörr. Please ensure that the full contributions of each author are acknowledged in the "Add/Edit/Remove Authors" section of our submission form.

https://journals.plos.org/plospathogens/s/submission-guidelines#loc-parts-of-a-submission

5) We notice that your supplementary Figures are included in the manuscript file. Please remove them and upload them with the file type 'Supporting Information'. Please ensure that each Supporting Information file has a legend listed in the manuscript after the references list.

Potential Copyright Issues:

i) Figure 8. Please confirm whether you drew the images / clip-art within the figure panels by hand. If you did not draw the images, please provide (a) a link to the source of the images or icons and their license / terms of use; or (b) written permission from the copyright holder to publish the images or icons under our CC BY 4.0 license. Alternatively, you may replace the images with open source alternatives. See these open source resources you may use to replace images / clip-art:

7) Please amend your detailed Financial Disclosure statement. This is published with the article. It must therefore be completed in full sentences and contain the exact wording you wish to be published.

8) Please send a completed 'Competing Interests' statement, including any COIs declared by your co-authors. If you have no competing interests to declare, please state "The authors have declared that no competing interests exist". Otherwise please declare all competing interests beginning with the statement "I have read the journal's policy and the authors of this manuscript have the following competing interests"

**Reviewers' Comments:**

Reviewer's Responses to Questions

**Part I - Summary**

Reviewer #1: Cross et al. examines the prevalence of carbapenem tolerance in a large panel of Enterobacterales bloodstream isolates. They show that Klebsiella pneumoniae strains exhibited the highest overall tolerance to the last line antibiotic meropenem. The authors employed an existing transposon library and elegant microscopy to uncover the mechanism of meropenem tolerance in K. pneumoniae. Their study links tolerance to envelope-stress systems (Rcs, PhoPQ, Cpx, OmpR/EnvZ) and identifies lytic transglycosylases as anti-tolerance factors, connecting survival to spheroplast formation and recovery. The mechanistic studies were well controlled with appropriate complemented strains in the supplementary materials.

A major strength is the high-throughput tolerance screen that still works despite MIC and growth-rate differences typically observed in panels of clinical isolates. Using a single fixed meropenem concentration (appropriate for time-dependent β-lactam killing) and quenching with a carbapenemase to avoid wash steps are innovative design choices. The screen’s rankings translated well to quantitative time-kill assays and will be of particular interest to the antibiotic tolerance community.

The manuscript would benefit from some additional discussion on if spheroplasts could contribute to carbapenem treatment failure in patients. Overall, this is an important study that shifts the focus beyond antibiotic resistance and highlights the need for understanding and targeting tolerance mechanisms.

Reviewer #2: In their manuscript entitled “Prevalence and mechanisms of high-level carbapenem antibiotic tolerance in clinical isolates of Klebsiella pneumoniae”, the authors characterize meropenem tolerance in clinical isolates in a range of species. Finding K. pneumoniae to be particularly tolerant to meropenem, subsequent work was focused on characterizing the genetic requirements for meropenem tolerance in K. pneumoniae. The work presented here provides molecular insight into the signaling pathways promoting tolerance in the presence of meropenem and potentially other beta-lactam antibiotics. I really appreciated the medium throughput assay for tolerance phenotypes in their collection of clinical K. pneumoniae isolates. The use of the carbapenemase to scavenge meropenem for the plating assay was very clever.

Altogether, the results suggest a collective and potentially sequentially activated stress response that is protective to K. pneumoniae grown in the presence of meropenem. At the same time, why and how this response leads to a halt in cell growth remains an open question, but an important one given the established connection between tolerance and slow growth at the population and single cell level. Is it possible that the cell envelope stress response preserves viability just long enough for cells to enter a slow growing, tolerant state? Finally, the rationale for some of the experiments, particularly the unusual use of the fluctuation test, is unclear and there are additional instances where more precise use of language and/or more clarity would be welcome.

Reviewer #3: The study by Cross et al. explores the prevalence and mechanisms behind carbapenem tolerance among clinical isolates of Klebsiella pneumoniae. Through screening of 271 bloodstream isolates Cross et all. identify higher prevalence of tolerance in K. pneumoniae isolates than in E. coli. A hyper-tolerant strain, TS1, survived nearly all meropenem exposure by forming cell wall‑deficient spheroplasts that recovered once the drug was removed. Using transposon sequencing, the authors identified cell envelope stress response systems (PhoPQ, Cpx, Rcs, OmpR/EnvZ) as essential for spheroplast stability and recovery, while the lytic transglycosylase MltB disrupted tolerance.

The area under scrutiny is clinically relevant and timely. Our understanding of the breadth of tolerance in clinically relevant context is important and the possibility that targeting stress response pathways or enzymes like MltB could aid eradication of tolerant infections is of broad interest.

**Part II – Major Issues: Key Experiments Required for Acceptance**

Reviewer #1: NA

Reviewer #2: 1. The fluctuation test is an unusual way to examine what appear to be non-heritable phenomena. Is the idea to look at variation in single cell responses? Overall, this section would be improved by revising for clarity as well as improving the presentation of the data. Specifically:

a. While there are details of the experimental method in the results sections, I didn’t see a methods section for this experiment. That should be included for reproducibility. How many wells were examined for each strain? Based on the graphs provided it looks like there are more replicates of the Δ4 in the 1 cell condition than WT.

b. Is OD600 a truly robust or meaningful readout for spheroplast populations when the Δ4 population doesn’t appear to readily generate spheroplasts (Fig 5C)?

c. There’s only one value on the Y-axis. Perhaps a supplemental figure could be included that show more clearly the areas between the meropenem added and when CV was calculated?

2. The authors need to address how their findings fit into the current understanding of factors underlying antibiotic tolerance, particularly with regard to growth rate. E.g. is expression of the stress response systems protective immediately following challenge with meropenem?

Reviewer #3: Tolerance assay:

It appears that only a single replicate is shown. Given that the rest of the manuscript relies on these observations, at least 3 biological replicates should be carried out and presented.

Fig 2B and 4C:

Please provide the number of cell frames imaged and information on biological replicates. Additional panels from the sample showing the prevalence of the observed phenotypes should be included in supplementary.

Statistical analysis for several figures is missing and should be included:

- Ln 120: statistical significance of the correlation is mentioned but not shown. I agree with the conclusion that biological significance is not present however given the analysis is mentioned, it should be included in the legend along with its results.

- Fig 6B: statistical analysis is missing, please correct.

- Fig 5: Please add details on replicates and statistics.

**Part III – Minor Issues: Editorial and Data Presentation Modifications**

Reviewer #1: 1. Presumably spheroplasts are very susceptible to other stresses. Can the authors comment on if these spheroplast could be a mechanism of tolerance that occurs during treatment of bloodstream infections.

2. Related: Are the tolerant K. pneumoniae spheroplasts still capsulated? If so, could the capsule provide protection from killing by whole blood?

3. Line 413-415: Interpretation of the fluctuation test results should be expanded to improve clarity. Does this result suggest that meropenem tolerant cells (spheroplasts) are pre-existing in this population before the addition of the drug? If so, could similar experiments be performed in more physiologically relevant conditions (serum or blood) to see if the trend holds up?

Reviewer #2: Major Comments:

1. The authors present TN-seq and mutant data to support a requirement for cell envelope stress response systems in meropenem tolerance. This work was performed in the TS1 strain, which exhibited higher tolerance to meropenem than E. coli MG1655 or another K. pneumoniae strain, KPNIH1. These cell envelope stress systems are broadly conserved in Gram-negative bacteria and not unique to TS1. How do these systems contribute to the underlying base-line differences in tolerance between organisms/strains? Do the author’s have any insights into why TS1 is more tolerant than other strains?

2. Is Meropenem special? The title suggests this phenomenon holds for all carbapenems but I don’t think this has been tested. If Meropenem is special or is the only drug tested the title should be revised to reflect this.

3. Do the authors have any thoughts as to why overexpression of rcsB seems to impair survivability in a WT background? (Fig 6 B).

4. The author’s find that loss of MltB and SltY, both of which should lead to accumulation of glycan strands, increase tolerance. I believe this is not unexpected result based on Cho et al 2014, who found that defects in the E. coli soluble lytic transglycosylase, SltA are protective in the presence of mecillinam, in accordance with their futile cycle model for beta-lactam killing in E. coli. This same paper indicates that a defect in mltB “imparts a fitness advantage in the presence of meropenem” which I believe similar to this study’s finding in K. pneumoniae if one assumes tolerance and resistance share overlapping mechanisms. Regardless, the relationship between this finding and those of Cho et al should be addressed in the results and/or discussion.

Minor Comments

1. Line 25: “novel” might not be the correct word. Most of these genes have been previously implicated in the response to cell envelope stress. Could just say identified cell envelope stress related factors as important for tolerance.

2. Line 40. Clarify that tolerance means “survive but do not grow”

Reviewer #3: Data/Analysis:

- Fig 3B: was a cut-off implemented for prioritizing genes for further validation?

- Ln 230-232: Given the validation hits seem to have been selected by picking hits that have exhibited low insertion rate after exposure – how were the two ‘survival supporting’ hits picked up? Surely those would have shown a complete opposite phenotype on the insertion rates? … I could find neither MltB or OsmB in table S4 so this cannot be extrapolated from there.

- Fig 4B: TS1 appears to afford almost the same survival as KPNIH1 mltB mutant - is the TS1 deficient or lacking the expression the mltB or is this phenotype arising from another variation?

- It is unclear why the stress responses are studied in TS1 but the prior section is done in KPNIH1, especially since the KPNIH library seems to exist – why were different mutants created in the TS1 background and the KPNIH library not exploited?

Figure/Table comments:

- Supplementary Table 1: Please clarify what this table is showing. Given Ln 143, this table appears to link to tolerance phenotypes, however its first mention at Ln65 which discusses resistance makes the data presented in this table difficult to place.

- Ln 163 and Table S2: MIC values are mismatched for TS1, please correct.

- Fig 3C: The y-axis is a little misleading. Are those fractions of survival of CFU per mL?

- Ln 400 and Fig 7: legend do not match. The figure implies that CV calculations were done at the blue dotted line timepoint while the main text suggests this was carried out at the red line timepoint. Please clarify.

Textual comments:

- Ln 64–65: Please reference for sentence starting “In some bacteria…”

- Ln 146: Please remove the stray “.” after “species”.

- Ln 244: Please define LTG.

- Ln 309: Please remove the aberrant “mutant” word.

- Ln 318: Please define CESR.

- Fig S4: appears in the text prior to FigS3, please correct the order.

- Ln 318-319 : ‘CESR single deletion mutants formed spheroplasts, albeit at much lower numbers compared to WT.’ is followed by a statement saying that 2/4 are exceptions. I would consider removing the above highlighted sentence and covering each deletion separately, especially given that half of the 4 mentioned are actually not what is described, essentially implying there is no single unique trend (beyond the overall tolerance effect)

- Ln 319-325: The structure of this paragraph is a little disjointed and hard to follow. I would suggest discussing all points on the rcs system first and then moving all the ompR/envZ.

- Ln 385: please remove ‘both’ when discussing 3 species.

- Throughout the manuscript hyphenation is often missing between ‘number(-)fold’.

PLOS authors have the option to publish the peer review history of their article (what does this mean? ). If published, this will include your full peer review and any attached files.

**Do you want your identity to be public for this peer review?** For information about this choice, including consent withdrawal, please see our Privacy Policy .

Reviewer #1: No

Reviewer #2: No

Reviewer #3: No

**Figure resubmission:**
---

## [Decision Letter · Decision Letter 1]

12 Jan 2026

PPATHOGENS-D-25-02804R1

Prevalence and mechanisms of high-level carbapenem antibiotic tolerance in clinical isolates of Klebsiella pneumoniae

PLOS Pathogens

Dear Dr. Dörr,

Thank you for submitting your manuscript to PLOS Pathogens. While the reviewers agree the current version of the manuscript is much improved, there are minor concerns to address. Therefore, we invite you to submit a revised version of the manuscript that addresses the points raised during the review process.

We look forward to receiving your revised manuscript.

Kind regards,

Renee M. Fleeman

Guest Editor

PLOS Pathogens

Matthew Wolfgang

Section Editor

PLOS Pathogens

Sumita Bhaduri-McIntosh

Editor-in-Chief

PLOS Pathogens

orcid.org/0000-0003-2946-9497

Michael Malim

Editor-in-Chief

PLOS Pathogens

orcid.org/0000-0002-7699-2064

**Journal Requirements:**

**Reviewers' Comments:**

Reviewer's Responses to Questions

**Part I - Summary**

Reviewer #1: All of my comments have been addressed

Reviewer #2: The authors have done an excellent job responding to reviewer comments.

Reviewer #3: I would like to thank the authors for their thoughtful revisions to the manuscript. Your updated explanations across the sections help place the findings in a broader context while the strengthened additional method details and improvements to the organization and legends, makes the study clearer and easier to follow. Overall, the revisions have significantly improved the manuscript, and I appreciate the care you put into addressing the feedback. I have no further concerns.

**Part II – Major Issues: Key Experiments Required for Acceptance**

Reviewer #1: All of my comments have been addressed

Reviewer #2: (No Response)

Reviewer #3: n/a

**Part III – Minor Issues: Editorial and Data Presentation Modifications**

Reviewer #1: All of my comments have been addressed

Reviewer #2: While I understand that meropenem is a representative carbapenem, the data do not support the idea that K. pneumoniae will respond to other carbapenems in the same way. To avoid confusion, I respectfully request title be revised to either specify that drug tested was meropenem or at least to indicate that only one carbapenem was tested. E.g. "Prevalence and mechanisms of high-level antibiotic tolerance to a representative carbapenem in clinical isolates of Klebsiella pneumoniae"

Reviewer #3: There is a lack of italicization on ln 40 for Klebsiella pneumoniae

PLOS authors have the option to publish the peer review history of their article (what does this mean? ). If published, this will include your full peer review and any attached files.

**Do you want your identity to be public for this peer review?** For information about this choice, including consent withdrawal, please see our Privacy Policy .

Reviewer #1: No

Reviewer #2: No

Reviewer #3: No

**Figure resubmission:**
---

## [Editor Report · Decision Letter 2]

20 Jan 2026

Dear Dr. Dörr,

We are pleased to inform you that your manuscript 'Prevalence and mechanisms of high-level carbapenem antibiotic tolerance in clinical isolates of Klebsiella pneumoniae' has been provisionally accepted for publication in PLOS Pathogens.

Best regards,

Renee M. Fleeman

Guest Editor

PLOS Pathogens

Matthew Wolfgang

Section Editor

PLOS Pathogens

Sumita Bhaduri-McIntosh

Editor-in-Chief

PLOS Pathogens

orcid.org/0000-0003-2946-9497

Michael Malim

Editor-in-Chief

PLOS Pathogens

orcid.org/0000-0002-7699-2064
---

## [Editor Report · Acceptance letter]

Dear Dr. Dörr,

We are delighted to inform you that your manuscript, "Prevalence and mechanisms of high-level carbapenem antibiotic tolerance in clinical isolates of Klebsiella pneumoniae," has been formally accepted for publication in PLOS Pathogens.

Best regards,

Sumita Bhaduri-McIntosh

Editor-in-Chief

PLOS Pathogens

orcid.org/0000-0003-2946-9497

Michael Malim

Editor-in-Chief

PLOS Pathogens

orcid.org/0000-0002-7699-2064